# Microbiological and Antioxidant Activity of Phenolic Compounds in Olive Leaf Extract

**DOI:** 10.3390/molecules25245946

**Published:** 2020-12-15

**Authors:** Dragana Borjan, Maja Leitgeb, Željko Knez, Maša Knez Hrnčič

**Affiliations:** 1Faculty of Chemistry and Chemical Engineering, University of Maribor, SI-2000 Maribor, Slovenia; dragana.borjan@um.si (D.B.); maja.leitgeb@um.si (M.L.); zeljko.knez@um.si (Ž.K.); 2Faculty of Medicine, University of Maribor, SI-2000 Maribor, Slovenia

**Keywords:** olive leaf extract, phenolic compounds, antimicrobial activity, antioxidant activity, enzymatic activity, supercritical fluid extraction

## Abstract

According to many reports, phenolic compounds isolated from olive leaves have very good biological activities, especially antimicrobial. Presently, the resistance of microorganisms to antibiotics is greater than ever. Therefore, there are numerous recent papers about alternative solutions for inhibiting their influence on human health. Olive leaf is studied as an important source of antimicrobials with low cost and used in medicine. Numerous publications on involving green technologies for isolation of active compounds from olive leaves have appeared over the past few decades. The present review reports on current knowledge of the most isolated phenolic compounds from olive leaf extract as well as methods for their isolation and characterization. This paper uses recent research findings with a wide range of study models to describe the antimicrobial potential of phenolic compounds. It also describes the vast range of information about methods for determination of antimicrobial potential focusing on effects on different microbes. Additionally, it serves to highlight the role of olive leaf extract as an antioxidants and presents methods for determination of antioxidant potential. Furthermore, it provides an overview of presence of enzymes. The significance of olive leaves as industrial and agricultural waste is emphasized by means of explaining their availability, therapeutic and nutritional effects, and research conducted on this field.

## 1. Introduction

Historically, olive leaves were used by old civilizations for the care of many illnesses. Ancient Egyptians used olive leaves for mummifying bodies of their pharaohs. In addition, later olive leaves became very popular folk remedy for fever. In the 1800s the British used them to handle tropical diseases such as malaria, which was so regnant in the colonies [1]. In the middle of the previous century, olive leaf extract was found to be positively acting on hypertension. Since then, research discussing olive leaves’ potentialities increased. Over the past few years, a lot of attention was paid to obtaining biologically active compounds from natural sources.

Olive (*Olea europaea* L.) is a woody oil tree that comes from Mediterranean region. It is famous for producing virgin olive oil. It has fusiform coriaceous grayish-green leaves (generally about 5–6 cm long and about 1–1.5 cm wide at mid-leaf) with fine edges and a short peduncle. The proportions may differ depending on tree age, culture conditions, production, and/or local pruning practice. The olive fruit is a drupe composed of the skin or epicarp, the pulp or mesocarp, and a woody shell holding the seed the pit or endocarp. The morphology of olives is not unique, but their chemical composition and organoleptic qualities are. Therefore, cardiologists and nutritionists consider their potential benefits for human health [2].

Olive leaf extract is a liquid with dark brown color and bitter in taste. Olive leaf extract is signified as a part of natural medicine with a wide range of health benefits. It has been used traditionally as an herbal supplement because it contains polyphenolic compounds with beneficial properties such as increasing energy levels, lowering blood pressure, and supporting the cardiovascular and immune systems. In addition to all beneficial effects mentioned above, olive leaf extract also has antimicrobial properties. As in the case of many natural products, the composition of the extract may vary according to different conditions, such as geographical location, cultivar, and plant nutrition [3].

Olive leaves (Figure 1) could also play effective roles in health care because they contain large amounts of other valuable phytochemicals, such as triterpenes, flavonoids, and chalcones [4]. Consequently, interest in the olive leaf and its chemical constituents has recently been raising. Polyphenols in olive leaves have been studied for various reasons, especially due to their visible antimicrobial properties.

The aim of this work is to review the antimicrobial effect of olive leaf extract against bacteria such *Escherichia coli*, *Bacillus cereus*, *Staphylococcus aureus*, *Pseudomonas aeruginosa*, *Candida albicans* and others. Numerous publications on isolation of active compounds from olive leaves have appeared over the past few decades. In this article, we review and present status of the current knowledge of the methods for isolation, characterization, presence of enzymes and determination of antimicrobial potentials of compounds found in olive leaves [6,7,8,9,10].

## 2. Main Chemical Compounds of Olive Leaves and Their Biological Effects

In the past few years, interest in obtaining biologically active compounds from natural sources is increased. Due to its phenolic content, the olive is one of the possible natural antimicrobial source.

Phenolic compounds are described as secondary metabolites that are derivatives of the pentose phosphate, shikimate, and phenylpropanoid pathways in plants. These compounds are one of the most widely occurring groups of phytochemicals. They are also of considerable physiological and morphological importance in plants. Structurally, despite their extreme variety, polyphenols possess a common carbon skeleton building block—the C6–C3 phenylpropanoid unit.

Olive leaves contain a large variety of phenolic derivatives and consist of:simple phenols (the most common and important low-molecular weight phenolic compounds);flavonoids (flavones, flavanones, flavonols, and flavan-3-ols);secoiridoids.

The structure of the main phenolic compounds identified in *Olea europaea* L. extract are shown in Figure 2.

Olive leaf extract contains many different compounds collectively termed olive biophenols, which are thought to give the extract its different therapeutic properties. The main component of all the constituents of olive leaf extract, oleuropein, has antimicrobial, antioxidative, antiviral, antiatherogenic, cardioprotective, and antihypertensive properties. Oleuropein belongs to a specific group of coumarin-like compounds, named secoiridoids. Oleaceas, Gentianales, Cornales, and many other plants are plentiful in this compound. Iridoids and secoiridoids are usually bound glycosidically. They are produced from the secondary metabolism of terpenes as precursors of different indole alkaloids [1]. Oleuropein, a (3,4-dihydroxyphenyl) ethanol (hydroxytyrosol) ester with α-glucosylated elenolic acid, is one of the iridoid monoterpenes. Oleuropein is considered to be responsible for the pharmacological effects. The main demonstrated biological activities of oleuropein are antioxidant and anti-inflammatory effects as well as the ability to treat oxidant and inflammatory-related diseases such as cardiovascular disease, hepatic disorder, obesity, and diabetes. Oleuropein shows anti-clastogenic activities, free-radical scavenging properties, and inhibits the development of different tumors cell types. This phenol can inhibit low-density lipoprotein oxidation and lipoxygenases. Furthermore, it has hypoglycemic and hypocholesterolemic activities. It is also known because of ability to improve lipid metabolism to facilitate obesity problems as described in different studies [11,12].

Under the action of light, high temperature, acid, or base oleuropein decomposes into hydroxytyrosol and elenolic acid [13]. Hydroxytyrosol is water- and lipid-soluble molecule, with the structure of a cathecol. It is present in the extra virgin olive oil in simple phenol form or esterified with elenolic acid. Hydroxytyrosol has a great impact against cardiovascular diseases and metabolic syndrome as well as on neuroprotection, antitumor, and chemo modulation effects; hence, it is a molecule of high interest to the pharmaceutical industry. This interest has led to wide research on its beneficial effects in humans and its biological activities as well as how to synthetize new molecules from hydroxytyrosol [14]. Brahmi et al. (2013) investigated and successfully tested hydroxytyrosol for free-radical scavenging activity [15].

Additionally, olive leaves contain a significant number of flavonoids, which represent the most common and widely distributed group of olive leaves polyphenols. Jemai et al. (2008) reported that they can be present in the aglycone form (apigenin, diosmetin, luteolin, quercetin,) or in the glycosylated form (luteolin-7-*O*-rutinoside, luteolin-7-*O*-glucoside, luteolin-5-*O*-glucoside, quercetin-7-*O*-rutinoside) [16].

Luteolin (3′,4′,5,7-tetrahydroxyflavone) has been identified as usually present in plants such as apple skins, broccoli, cabbages, carrots, celery, chamomile tea, green peppers, olive, onion leaves, and perilla leaf, and it is a significant member of the flavonoid family [17]. Plants with a high content of luteolin have been used ethnopharmacologically to treat inflammation-related symptoms. This flavonoid inhibits chromosome alterations and presents antioxidant effects, anti-tumorigenic properties, and anti-inflammatory activities. Additionally, luteolin potentially controls colon cancer in multiple aspects [18].

After all, using the extract without isolating the constituents might be recommended to achieve health because of the synergistic effects of all phenolic compounds present in the extract. Rodriguez et al. (2007) studied one of the most important synergistic effects of phenols from olive. They found that results could suggest a possible synergistic effect between hydroxytyrosol and 3,4-dihydroxyphenylglycol [19]. Also, Rubio-Senent et al. (2015) found that synergistic effect occurred when the hydroxytyrosol acetate and the 3,4-dihydroxyphenylglycol were supplemented by hydroxytyrosol [20]. Also, the health-promoting properties of oleuropein and other polyphenols from olive leaves are widely researched in the literature.

### Antimicrobial and Enzymatic Activity of Phenolic Compounds

Presently, the resistance of microorganisms to antibiotics is greater than ever. There are numerous recent reports about alternative solutions for inhibiting their influence on human health. According to many reports, phenolic compounds isolated from olive leaves have very good biological activities [9,21,22,23], especially antimicrobial [24,25,26,27].

*Escherichia coli* is a group of bacteria that usually lives in the intestine of humans and animals and helps to keep our guts healthy. However, according to the Centers for Disease Control and Prevention certain types of the bacteria can occasionally cause severe illness. Pathogenic strains of *Escherichia coli* can be ingested with contaminated food, such as undercooked ground beef, soft cheeses made from raw milk, fresh products, grains, or even contaminated beverages, including water, unpasteurized milk, and fruit juices. Even though *Escherichia coli* can infect anyone specified groups of people including children, older adults and people with weakened immune systems are more at risk of developing symptoms than others. It has been published a wide range of studies regarding the great antimicrobial activity against *Escherichia coli* of polyphenols, such as oleuropein, rutin, hydroxytyrosol, and caffeic acid [3,6,7,24,25,26,28,29,30].

Salmonella bacteria usually live in animal and human intestines and are shed through feces. Humans become infected most often through contaminated water or food. Typically, salmonella infection does not cause any specific symptoms. However, some infected people develop symptoms such as diarrhea, fever, and abdominal cramps within eight to 72 h. Most healthy people recover after few days without specific treatment or medication. Medical attention is required sometimes due to dehydration caused by diarrhea associated with salmonella infection. *Salmonella enteritidis* can be inside perfectly normal-appearing eggs and can cause illness if the eggs are eaten raw or undercooked. According to Liu et al. (2017) oleuropein and verbascoside have great impact against *Salmonella enteritidis* [30]. Also, Erdohan et al. (2011) have recently investigated apigenin-7-glucoside, caffeic acid, catechin, diosmetin-7-glucoside, hydroxytyrosol, luteolin, luteolin-7-glucoside, oleuropein, vanilic acid, verbascoside, rutin, and tyrosol against *Salmonella enteritidis* [7]. On the other hand, recent studies have reported antimicrobial activity of the olive leaf extract against *Salmonella typhimurium* [7,29].

Staph infections are caused by Staphylococcus bacteria, types of germs ordinarily present on the skin or in the nose of even healthy people. Most of the time, these bacteria cause no problems with exception of resulting in relatively minor skin infections. However, staph infections can turn deadly if the bacteria invade deeper into human body, entering bloodstream, bones, heart, joints, or lungs. Number of otherwise healthy people who are developing life-threatening staph infections is increased. Antibiotics and drainage of the infected area are mostly used for treatment. However, some staph infections no longer respond to common antibiotics. Latterly, a lot of scientific interest has been aimed at *Staphylococcus aureus*, *Staphylococcus capitis*, *Staphylococcus epidermidis*, *Staphylococcus hominis*, *Staphylococcus pyogenes, Staphylococcus xylosus* [3]. *Staphylococcus aureus* can cause a wide range of illnesses. It can be a cause of various skin infections, such as abscesses, boils, carbuncles, cellulitis, folliculitis, impetigo, pimples, and scalded skin syndrome. Also it can bring about some life-threatening diseases such as endocarditis, meningitis, osteomyelitis, pneumonia, sepsis, and toxic shock syndrome. It is one of the most usual causes of hospital-acquired infections and is generally the cause of wound infections following surgery. Many reports have shown that caffeic acid, oleuropein, rutin, and verbascoside are suitable choice against *Staphylococcus aureus* [6,7,22,24,25,26,27,28,29].

Bacillus species are gram-positive bacteria which are facultatively anaerobic, endospore-forming aerobic or rod-shaped. Later with age, in some species cultures may turn gram-negative. Many species of the genus live in every natural environment because of exhibition a wide range of physiologic abilities. Only one endospore can be formed per cell. The spores are resistant to cold, heat, desiccation, disinfectants, and radiation. *Bacillus cereus* is a gram-positive, spore-bearing rod that is widely distributed in the environment where spores persist under adverse conditions and can grow when readily decomposable matter is available. This bacterium produce toxin and is one of the most common causes of food poisoning, which is called “fried rice syndrome.” According to recent research, oleuropein, apigenin-7-*O*-glucoside, caffeic acid, luteolin 4′-*O*-glucoside, luteolin 7-*O*-glucoside, rutin, and verbascoside have great synergy effect against *Bacillus cereus* [3,6,29]. Additionally, recent studies have reported that chemical composition of the olive leaves is great against *Bacillus subtilis*, one of the best-characterized bacterium among gram-positives [3,6,28].

*Klebsiella Pneumoniae* is a gram-negative bacterium. Even though it is present in the normal flora of the mouth, skin, and intestine, it can be the cause of damaging changes to human and animal lung if aspirated, especially to the alveoli resulting in bloody sputum. Many researchers determined that oleuropein, rutin, and hydroxytyrosol have great impact against *Klebsiella Pneumoniae* [3,6,28,29].

Bacteria and other microorganisms, such as fungi, make a wide range of problems in many areas from nutrition to health care and different implants. Pereira et al. (2007) are shown that constituents of olive leaf extract, such as apigenin 7-*O*-glucoside, caffeic acid, oleuropein, luteolin 7-*O*-glucoside, luteolin 4′-*O*-glucoside, rutin, and verbascoside had protective effects in fungi *Cryptococcus neoformans* [6]. *Cryptococcus neoformans* is a fungus which is present world widely in the environment. People normally do not get sick of it but if infection occurs the cause is the inhalation of microspores of *Cryptococcus neoformans. Cryptococcus neoformans* infections are infrequent in people who are otherwise healthy. Most cases happen in people who have a weakened immune systems, especially those who are in the advanced stages of HIV/AIDS.

*Candida* species are the most usual infectious fungal species in humans and about 150 is known. *Candida albicans* is most widespread pathogenic species, largely influencing on immunocompromised individuals. Beside its role as the primary etiology for different types of candidiasis, *Candida albicans* is famous for contributing to polymicrobial infections. Despite being a normal part of the human microflora or the microorganisms that commonly live in or on human bodies, *Candida albicans* can cause infections if it overgrows. Additionally, *Candida* species can infect mucus membranes and skin. It was announced that oleuropein and hydoxytyrosol are the best inhibitors [24,27,28]. Despite antimicrobial activity of olive leaf extract against *Candida albicans*, it was reported that oleuropein has a great impact on suppression of *Candida glabrata* and *Candida parapsilosis* [3].

## 3. Methods of Isolation, Characterization, and Determination of Antimicrobial Potentials of Main Chemical Compounds in Olive Leaves

### 3.1. Isolation of Main Chemical Compounds from Olive Leaves

Bioactive compounds from natural sources are possibly applicable in food, chemical, and pharmaceutical industries; hence the interest in the development of bioprocesses for the production or extraction of these compounds has increased in recent years. The nature of the target compounds present in the crude extracts or fractions is the most significant factor that should be considered before planning of isolation process. The general characteristics of the molecules to take into account involve:acid-base properties;charge;molecular size;solubility (hydrophobicity or hydrophilicity);stability.

#### 3.1.1. Traditional Extraction Techniques

One of the most important steps in sample pre-treatment for polyphenols analysis is extraction. In general, extraction is a separation process where the distribution of the analyte (in this case phenolic compounds) among two phases is made to arrive at the appropriate distribution coefficient. Due to the disease-preventing effects and special health-promoting of polyphenols, efficient methods for extracting polyphenols from various plants have been investigated and evolved [31]. Some of the most frequently used techniques are described below.

##### Solid-Liquid Extraction Technique (SLE)

Solid-liquid extraction by maceration of the olive leaves in a solvent is the most often used extraction system. Solid-liquid extraction with organic solvents is extensively used for the isolation of analytes from a solid matrix for industrial needs. Interest in usage of different solid-liquid extraction methods for extraction of intracellular compounds and liquids from plant cellular tissues in commercial purposes rapidly increased over the past decade. Furthermore, separation of phenols (especially oleuropein) from olive leaves based on traditional solid-liquid extraction methodology has been the subject of a few patents. On a laboratory scale, phenols are most frequently isolated by using traditional methods such as maceration and Soxhlet extraction with different extractants (methanol-water mixtures or hexane). However, the use of these solvents could have hazardous effect on the human health and the residues of the solvents may also remain in the final products [32].Therefore, if the procedure is on industrial scale and, especially if the products are targeted for human use, it is necessary to avoid toxic extractants. Extraction time is mostly in the range from 24 h to 48 h and it is desirable to shorten it as much as possible.

Solid-liquid extraction for extracting bioactive compounds from olive leaves was investigated by many researchers. According to Rahmanian et al. (2015) this technique can be upgraded by various treatments such as steam blanching or acid hydroxylation for extracting useful compounds [33]. For recovery of bioactive compounds are commonly used Soxhlet and solid–liquid extraction by direct contact between the raw material and the solvent, with or without agitation. Even though these technologies ensure a fast and reproducible extraction, large amount of organic solvent is used. Usual extraction solvents used for polyphenols extraction from olive leaves are acetone, methanol, ethanol, n-hexane, ethyl acetate, chloroform, and diethyl ether, as well as aqueous alcohol mixtures. The selection of extraction solvent is an important criterion for extracting desired components from plant materials such as dissolving power, selectivity, volatility, and cost. Furthermore, extracting solvents have an extraordinary influence on the antimicrobial properties of olive leaves.

Khalil et al. (2013) investigated the influence of extract concentration, contact time, pH, and temperature on the reaction rate and the shape of the Ag nanoparticles, which were synthesized using hot water olive leaf extracts as reducing and stabilizing agent and evaluated for antibacterial activity against drug resistant bacterial isolates [22]. Lee-Huang et al. (2003) published a study about antiviral activity of olive leaf extract preparations, standardized by liquid chromatography-coupled mass spectrometry, against HIV-1 infection and replication, where water was also employed as a solvent during extraction process [23]. Furthermore, Goulas et al. (2009) used olive leave water extracts against endothelial cells and cancer [34].

Jafari et al. (2017) showed how to incorporate microencapsulated olive leaf methanolic extract into tomato paste to get benefit from antimicrobial properties of the extract against *Aspergillus flavus* over short and long-time storage [35]. On the other hand, Al-Quraishy et al. (2017) employed olive leaf methanolic extract for investigation of preventing hydrochloric acid/ethanol-induced gastritis in rats by attenuating inflammation and augmenting antioxidant enzyme activities [36]. Additionally, methanol as a solvent was used for extraction of olive leaves in investigation for potential anti-tumoral activities; in this study specifically human breast cancer cells were treated [37].

Čabarkapa et al. (2014) used dry olive leaf ethanolic extract in adrenaline induced DNA damage evaluated using in vitro comet assay with human peripheral leukocytes [38]. Additionally, Poudyal et al. (2010) employed solid-liquid extraction with ethanol in exploring how phenol-enriched extract, with oleuropein as the main component, attenuated the cardiovascular, hepatic, and metabolic signs of a high-carbohydrate, high-fat diet in rats [39]. On the other hand, Liu et al. (2017) investigated the antimicrobial effect of olive leaf ethanolic extract against major foodborne pathogens, including *Escherichia coli*, *Listeria monocytogenes*, and *Salmonella enteritidis* and demonstrated a sufficient concentration of ethanolic extract for almost completely inhibition of growth of these three pathogens [30].

Paiva-Martins et al. (2017) studied the possibility of preparing olive oil with comparable stability characteristics and nutritional value to virgin olive oil. They enriched refined olive oil with polyphenols extracted from the leaves of several olive cultivars using n-hexane [40].

Yuan et al. (2015) demonstrated a high antioxidant activity of the ethyl acetate extract enzymatic hydrolysate, their results suggested that hemicellulase has promising and attractive characteristics for industrial production of hydroxytyrosol. Also, they indicated that hydroxytyrosol could be a valuable biological component for use in pharmaceutical products and functional foods [13]. On the other hand, Turkez et al. (2012) used olive leaf extract obtained by solid-liquid extraction with ethyl acetate to determine the effectiveness of olive leaf extract in modulating the permethrin induced genotoxic and oxidative damage in rats [41].

Altıok et al. (2008) published a study about the isolation of polyphenols from the extracts of olive leaves by adsorption on silk fibroin, where acetone was employed as solvent in extraction process and silk fibroin was used as a novel adsorbent to recover the polyphenols from the olive leaf extract [42]. Additionally, Karygianni et al. (2014) used solid-liquid extraction with acetone for investigating the antimicrobial activity of the olive leaf extract against different oral microorganisms (ten bacteria and one fungus strain) [25].

Erdohan and Turhan [7] dealt with olive leaf extract obtained by solid-liquid extraction with chloroform and used for antimicrobial food packaging. Active packaging prolongs shelf life and keeps safety and quality of food products by extending the lag phase and reducing the growth rate of food pathogens or spoilage microorganisms [7].

Even though the organic solvents provide the extraction of the metabolites from plants, additional purification can be essential in order to obtain concentrated specific components selectively because many other compounds such as metals, proteins, or sugars may be present in the plant extracts.

##### Soxhlet Extraction Technique

Soxhlet extraction includes repeated solvent distillation through a solid sample to take away desired analyte. The sample is placed in a thimble-holder and during operation is filled with condensed fresh solvent from a distillation flask step by step. When the liquid achieves an overflow level, a siphon aspirates the whole contents of the thimble-holder and unloads it back into the distillation flask, carrying the extracted analytes in the bulk liquid. When complete extraction is reached repetition of this operation is stopped. This property represents Soxhlet as a hybrid continuous–discontinuous technique. However, the system has a continuous character since the solvent is recirculated through the sample. Soxhlet extraction is a very simple technique and it makes the procedures for different samples very similar.

Soxhlet extraction has many advantages and it was confirmed more than a century ago. Additionally, most of drawbacks have been overcome in the following ways:by assisting extraction with auxiliary energies;by automating extraction with different approaches;by increasing the pressure in the sample cartridge.

Luque de Castro et al. (2010) published that prosperity of the conventional extractor by incorporation of the commonly technologies, such as the automation of Soxhlet extraction, is performed on the commercial equipment, which ensures substantial savings in time and extractant [43].

Soxhlet extraction technique has some disadvantages that include:lack of versatility;limited solvent choice;long extraction time;possible degradations of the target compounds due to local overheating;relatively high costs due to solvent consumption.

Because of the economical disadvantages of Soxhlet technique, it is less used in research and only on a small scale.

According to results of Şahin et al. (2011) Soxhlet extraction of olive leaves is a good way to extract oleuropein; different types of solvents (water, ethanol, methanol, hexane, and methanol/hexane mixture) were used to define the effect of the solvent type on the extraction performance [11]. In another study, Soxhlet extraction has been used to get mixture of phenolic compounds oleuropein, luteolin-7-*O*-glucoside, luteolin-4′-*O*-glucoside, luteolin, and hydroxytyrosol acetate. They concluded that olive leaf extract prevents lead (Pb)-induced abnormalities in behavior and neurotransmitters production in chronic Pb exposure in rats. The aim of their study was to provide additional evidence that olive leaf extract acts as an anti-apoptotic, anti-inflammatory, and antioxidant mediator in Pb exposed rats [44].

#### 3.1.2. Non-Conventional Extraction Techniques

Presently, it is necessary to shorten the extraction time and reduce the costs with lower solvent consumption; therefore, the application of alternative was extraction and isolation techniques is becoming more and more widespread. They include:microwave-assisted extraction;ultrasound-assisted extraction;supercritical fluid extraction (the most used are subcritical water extraction and supercritical carbon dioxide extraction).

However, recent studies showed that the application of thermal treatments expand in olive oil industry. One of researched is steam explosion, which is performed at high pressures and temperatures with saturated steam for a few minutes and followed by an explosive decompression [45]. Despite its many advantages, this thermal treatment has not been further developed for industrial application due to technical complexity of explosive decompression [46]. The other thermal treatment is discontinuous steam treatment which is performed at lower pressures and temperatures. Explosive decompression is not necessary so that this thermal treatment was scaled up for industrial application [46].

Valavanidis et al. (2004) compared thermal stability of olive oil under thermal treatment at elevated temperatures and concluded that it has higher resistance to different chemical changes than other vegetable oils. They studied degradation of polyphenols under heating and found that concentrations of oleuropein and hydroxytyrosol rapidly decrease with time [47].

##### Microwave-Assisted Extraction (MAE)

MAE technique involves lower consumption of organic solvents and reduced costs [48]. MAE provides a significant shortening of the leaching time as compared to the conventional procedure of maceration that usually requires so it has been used as an alternative extraction method on a laboratory scale [49].

The heating using microwaves is based on direct effect of microwaves on molecule via two mechanisms, specifically, by ionic conduction and dipole rotation [50,51]. Factors which influence on extraction process are characteristics of matrix, extraction time and temperature, microwave power, and solvent volume and type [52]. It is supposed that microwave-assisted extraction process involves three following steps:separation of the solutes from the active sites of the sample matrix under increased pressure and temperature;diffusion of solvent across the matrix of sample;release of the solutes from the sample matrix to the solvent.

Compared to the conventional techniques (e.g., maceration) the MAE has many advantages, for instance:high extraction selectivity, which make it a desirable technique in extraction of phenolic compounds from olive leaves;higher extraction efficiency;less working time;shortened extraction time.

It also provides higher recoveries compared to traditional extraction methods (e.g., Soxhlet extraction), therefore MAE has acquired much attention for phenolic compounds extraction from plants and, particularly, from olive leaves. [8,53] Phenolic compounds can be simply extracted by MAE. However, MAE requires higher temperatures (110–150 °C) as Grigonis et al. (2005) published in their study [54].

Mohammadi et al. (2015) employed MAE to obtain olive leaf extracts encapsulated by nano-emulsions in soybean oil for evaluating the antioxidant activity [12]. MAE was also used in a pilot study on the DNA-protective, cytotoxic, and apoptosis-inducing properties of olive leaf extracts. Results of this study could help with explanation of beneficial effect of olive leaf extract and further investigation of its mechanism of action may lead to great progress in the prevention of human cancer [55].

Taamalli et al. (2011) developed a simple and fast method for the extraction of phenolic compounds from olive leaves. They compared MAE to conventional methods. According to efficiency and speed they concluded that MAE can be a useful alternative for the characterization of phenolic compounds from olive leaves. Solvent type and composition, extraction time, and microwave temperature are experimental variables which influence on MAE process and they were optimized by univariate method [56].

##### Ultrasound-Assisted Extraction (UAE)

Ultrasound causes cavitation depends on the characteristics of ultrasound, product properties and ambient conditions. UAE requires an energy generator, a liquid medium, and a transducer [57]. Ultrasound improves the extraction rate by increasing the mass transfer rates and potential rupture of cell wall because of the formation of micro cavities, leading to higher product yields with reduced processing time, solvent consumption, thermal degradation losses, water and energy consumption [58,59]. The uses of ultrasound are broadly distinguished into two groups: high intensity and low intensity.

Two types of devices can be used for application of high power ultrasound: ultrasonic bath and probe-type ultrasound equipment.

Both systems are based on a transducer as a source of ultrasound power. The most usual type used is the piezoelectric transducer. The ultrasonic bath consists of a stainless-steel tank with one or more ultrasonic transducers. It ordinarily operates at a frequency around 40 kHz. Additionally, it can be equipped with temperature control. It is available and readily cheap with possibility of simultaneously treatment of large number of samples. However, reproducibility and power of ultrasound delivered directly to the sample are low. The delivered intensity is highly weakened by the water contained in the bath and the glassware used for the experiment. On the other hand, the probe system is more powerful due to an ultrasonic intensity delivered through a smaller surface. It usually operates at around 20 kHz and uses transducer bonded to probe which is immersed into the reactor resulting in a direct delivery of ultrasound in the extraction media with minimal ultrasonic energy loss [60]. 

Because of possible economic gains developments in ultrasonic equipment are considered for commercial opportunities. However, the implementation of UAE on the industrial scale has still been a challenge. It was proved that UAE is faster and more efficient compared to conventional extraction in olive leaves. The explanation lies in the fact that this method is a powerful aid in accelerating different steps of the analytical process. The major advantages of UAE are:low-temperature levels;high yields;short process time.

Şahin et al. (2013) used UAE to obtain polyphenols from agricultural and industrial waste of olive oil and table oil productions. Their study contributed to the simulation and optimization of conditions of olive leaf extraction [10].

##### Supercritical Fluid Extraction (SFE)

SFE is a novel method for extracting high added value compounds in low quantities from solid plant matrixes [61,62]. Separation of compounds that are low in volatility and are thermally labile has been primary reason for development SFE as an alternative method. However, with increasing public interest in natural products, supercritical fluid extraction may become a regular extraction technique for studying agricultural samples, food, and herbal. Besides short extraction time (being rapid and automatable) and high efficiency the most advantages of SFE consist of:less solvent consuming (environmentally friendly);very cleaner extracts;selectivity of the extraction, which can be manipulated by changing pressure and/or temperature [11,63]extraction of nonpolar compounds by this procedure has very low energy costs.

However, like many other processes, SFE is sometimes criticized for its large number of factors, which need to be properly adjusted before every single run.

As solvents are usually used sub- or supercritical fluids, such as water and carbon dioxide. The intrinsic low viscosity and high diffusivity of supercritical CO_2_ has permitted relatively clean extracts [11,64]. Furthermore, the absence of light and air during extraction reduces the degradation of analytes, conversely to traditional extraction techniques [65].

The solubility depends on the solvent density that may vary notably with changing extraction conditions, especially when operating in the sub- or supercritical region.

Water, a cheap and environmentally friendly solvent is an ideal solvent for industrial extraction of phenolics but, its use is limited due to poor extraction efficiency at low temperatures [66]. Subcritical water refers to water at temperature between 100 °C and 374 °C and pressure < 221 bar [67], which is high enough to keep the liquid state (below the critical pressure). Although the literature regarding the use of subcritical water extraction for obtaining bioactive compounds from olive leaves was found to be scarce, it surely has a great potential for olive leaf extraction soon [68].

SFE with carbon dioxide is also very attractive because of safety of solvent, which is non-toxic and non-flammable as well as odorless, colorless, and tasteless. Furthermore, carbon dioxide as a solvent is highly selective, easily removable, readily available, and cheap. Also, great advantages is that the extraction parameters can be changed in a wide range of pressure and temperature [69].

Higher investment cost as for traditional extraction techniques are needed and it is the only serious drawback of supercritical carbon dioxide extraction. On the other hand, the operating costs are relatively low. Furthermore, it is very simple to be scaled up to the industrial scale.

Jimenez et al. (2011) published a study about the oxidative stability of oils containing olive leaf extracts obtained by supercritical carbon dioxide extraction with ethanol as cosolvent at temperature of 40 °C, under pressure of 300 bar and at constant flowrate of carbon dioxide for 5 h [64].

Şahin et al. (2011) studied oleuropein content in olive leaf extract obtained by supercritical carbon dioxide extraction. They compared these results with results obtained by Soxhlet method. The effect of pressure in the range from 100 bar to 300 bar, temperature from 50 °C to 100 °C, and type of cosolvent in the amount of both extract and oleuropein were investigated [11].

### 3.2. Analytical Methods

The identification of the phenolic components of olive leaves has attracted interest in the last few years owing to their biological and pharmacological activity. Firstly, the total content of phenolic compounds was determined by spectrophotometric techniques. However, the only chance to identify and quantify the polyphenols present in olive leaves singularly is a previous separation of the compounds before their detection.

Gas chromatography (GC) is typically used for separating and analyzing compounds which can be vaporized without decomposition. The use of GC has been reported in many studies as possible technique for phenolic characterization in olive leaves [26,70,71].

The complete process of mass spectrometry (MS) includes the conversion of the sample into gaseous ions, with or without fragmentation, which are then characterized by their mass to charge ratios and relative abundances. De Nino et al. (1997) studied direct identification of phenolic compounds, especially oleuropein, from olive leaf extracts using MS [70].

In a gas GC/MS system, the MS scans the masses continuously throughout the separation. Salta et al. (2007) described this method in their scientific paper. A GC/MS method was applied for detection of three triterpenoic acids and 25 target polyphenolic compounds, such as cinnamic acid, ferulic acid, hydroxytyrosol, *p*-hydroxy-benzoic acid, *p*-hydroxy-phenylacetic acid, protocatechuic acid, *p*-coumaric acid, tyrosol, vanillin, and vanillic acid [71].

Liquid chromatography/mass spectrometry (LC/MS) combines the resolving power of LC with the detection specificity of MS. LC/MS may be applied in a wide range of sectors so that was described in many scientific articles [34,37,64,72,73,74,75,76]. Lee-Huang et al. (2003) prepared and standardized olive leaf extract using liquid chromatography/mass spectrometry and tested these olive leaf extract preparations for activity against HIV-1 infection and replication, where pure oleuropein was used as the standard and several peaks were detected corresponding to oleuropein, olenolic acid, and hydroxytyrosol [23].

Nuclear magnetic resonance (NMR) spectroscopy is an analytical chemistry technique used for determination of sample content and purity. NMR is applied in the analysis of complex mixtures without previous separation of the individual components in the mixture. The technique can be used for studying physical properties at the molecular level such as conformational exchange, diffusion, phase changes, and solubility. This was described by Kontogianni et al. (2013) who provided quantitative results of dry olive leaf extract for following compounds: hydroxytyrosol, luteolin-4′-*O*-β-d glucopyranoside, luteolin-7-*O*-β-d-glucopyranoside, luteolin, and oleuropein [73].

High performance liquid chromatography (HPLC) has a great versatility, and it is widespread in science and industry as describes in many studies [10,56,72,74,76,77]. Jemai and co-workers employed HPLC analysis for identifying oleuropein and oleuropein aglicone in extract and hydroxytyrosol as a major compound and confirmed by LC/MS [16]. The following phenolic compounds, apigenin-7-glucoside, flavonoid x, luteolin-7-glucoside, oleuropein, oleuroside, rutin, and verbascoside, in olive leaf extract were identified and characterized using HPLC method with photodiode array detector in investigation of anti-HIV activity [23]. Liu et al. (2017) investigated the antimicrobial effect of olive leaf extract against major foodborne pathogens, including *Listeria monocytogenes*, *Escherichia coli* O157:H7, and *Salmonella enteritidis* and employed HPLC for identifying four major peaks in olive leaf extract: luteolin-7-*O*-glucoside, luteolin-4-*O*-glucoside, oleuropein, and vabascoside [30]. Altıok et al. (2008) isolated the polyphenols from olive leaves and investigated the influence of extraction conditions on the total phenol content and antioxidant activity of olive leaf extract, and HPLC analysis with diode array detector was used for the quantification of oleuropein and rutin [42]. Abaza et al. (2007) investigated protective effects of olive leaves by incubating human promyelocytic leukemia cells with the olive leaf extracts from seven principal Tunisian varieties and employed HPLC method with diode array UV detector for determining a few minor compounds present in olive leaf extracts, where two main compounds were apigenin 7-glucoside and oleuropein. This compounds were present in all extracts and at relative high concentrations [78]. Flemmig et al. (2011) published a study which provides a rational basis for the traditional use of olive leaves against gout in Mediterranean folk medicine, where they identified hydroxytyrosol, tyrosol, oleuropein, verbascosid, luteolin-7-*O*-β-d-glucoside, apigenin-7-*O*-β-d-glucoside using HPLC method with UV-DAD detector [79]. Al-Quarishy et al. (2017) researched if olive leaf extract could protect gastric mucosa against HCl/ethanol-induced gastric mucosal damage in rats and used HPLC with UV-VIS detector for detection of the following compounds from olive leaf extract: tyrosol and its derivative, oleuropein, caffeic acid, rutin, luteolin derivatives, and vanillin [36]. Additionally, group of researchers investigated protective activity of the olive leaf extract in gastric mucosal injury induced by a corrosive concentration of ethanol. Using HPLC they determined the major constituent of the olive leaf extract was oleuropein, composing 19.8% of the extract, the other identified compounds were: caffeic acid, vanillin, rutin, luteolin, luteolin-7-*O*-glucoside, apigenin, apigenin-7-*O*-glucoside, quercetin, and chryseriol [80]. Salta et al. (2007) published a study about oxidative stability of edible vegetable oils enriched in polyphenols with olive leaf extract where identification and quantification of oleuropein was performed by HPLC analysis with diode array and fluorescence detector [71]. Separation and quantification by HPLC method with UV-VIS detector of caffeic acid, hydoxytyrosol, tyrosol, and oleuropein were reported in study about the neuroprotective effect of olive leaf extract and its relation to improvement of blood–brain barrier permeability and brain edema in rat with experimental focal cerebral ischemia [81]. Pereira and co-workers employed HPLC system with diode array detector to identify following phenolic compounds: apigenin 7-*O*-glucoside, caffeic acid, luteolin 7-*O*-glucoside, rutin, luteolin 4′-*O*-glucoside, oleuropein, verbascoside, [6]. Karygianni et al. (2014) performed the analysis of standard solutions of oleuropein and hydroxytyrosol by this method [25]. Additionally, Mylonaki et al. (2008) used HPLC system with diode array detector analysis on olive leaf extracts and got peaks for next compounds: apigenin 7-*O*-rutinoside, luteolin 3′-*O*-glucoside, luteolin 3′,7-*O*-diglucoside, luteolin 7-*O*-glucoside, luteolin 7-*O*-rutinoside, oleuropein, quercetin 3-*O*-rutinoside (rutin) [75]. Japón-Luján et al. (2008) acquired the chromatograms for apigenin-7-glucoside, luteolin-7-glucoside, oleuropein, and verbascoside, in their study [82].

### 3.3. Methods for Determination of Antimicrobial Potential

Polyphenols have multiple biological effects, including antimicrobial activity. The precise antimicrobial mechanism of polyphenols in olive leaf extract is unknown. However, it has shown antibacterial activity that slows the growth rate of microorganisms, inhibits several enzymes, micrococcal nuclease, lysozyme, and causes damage to cell membranes [26].

Generally, the antimicrobial capacity of phenolic compounds is well-investigated. Both individual phenolics and their mixtures were tested for antimicrobial activity.

Antimicrobial tests were performed to determine the effect of olive leaf extract against the growth of certain bacteria. Baycin et al. (2007) were exploring the antibacterial action of oleuropein and rutin against four bacteria (*Escherichia coli*, *Staphylococcus aureus*, *Klebsiella pneumoniae*, and *Pseudomonas aeruginosa*) determined by the disk diffusion method, where bacterial cultures were grown on agar overnight and after incubation period the width of the inhibition zone was measured [24].

Sudjana et al. (2008) determined the antimicrobial activity of the extract for many organisms using the broth microdilution assay. Minimum bacterial concentrations (MBCs) and minimum fungicidal concentrations (MFCs) were determined from broth microdilution assays. MBCs/MFCs were defined as the lowest concentration killing ≥ 99.9% of the inoculum compared with initial viable counts. Assays were repeated at least three times. Modal minimum inhibitory concentration (MIC) and MBC/MFC values were chosen. Subculture as the lowest concentration resulting in the maintenance of, or a reduction in, the number of organisms in the inoculum was used for determination of MICs. It was concluded that olive leaf extract is most active against *Campylobacter jejuni*, *Helicobacter pylori*, and *Staphylococcus aureus* [3].

Regular well agar diffusion method was carried out to detect the activity against clinical bacterial. For antibacterial activities of the compounds, wells were made in plates containing nutrient agar with each clinical isolate. The silver nanoparticles were synthesized using hot water olive leaf extracts, as reducing and stabilizing agents are reported and evaluated for antibacterial activity against drug resistant bacterial isolated by Khalil et al. (2013). They revealed significantly inhibited growth against multi drug resistant *Staphylococcus aureus*, *Pseudomonas aeruginosa*, and *Escherichia coli* using the standard well agar diffusion method [22].

### 3.4. Methods for Determination of Antioxidant Activity

Another important property of olive leaf extracts is their antioxidant activity, which is confirmed in many research articles. There are developed many methods for analyzing antioxidant activity. Some of the most frequently used methods are described below.

#### 3.4.1. N,N-Dimethyl-p-phenylenediamine Dihydrochloride (DMPD) Method

The principle of the assay is that at an acidic pH and in the presence of a suitable oxidant solution N,N-Dimethyl-*p*-phenylenediamine dihydrochloride can form a stable and colored radical cation (DMPD^•+^) (Figure 3, step 1). Antioxidant compounds which can transfer a hydrogen atom to DMPD^•+^ quench the color and produce a discoloration of the solution which is proportional to their amount (Figure 3, step 2). This reaction is fast, it takes less than 10 min. As a measure of the antioxidative efficiency is taken the end point, which is stable. Therefore, this assay reflects the capability of radical hydrogen-donor to scavenge the single electron from DMPD^•+^.

Briante and co-workers used this method in their research about what they published several studies [84,85]. They obtained a large quantity of highly purified hydroxytyrosol in short time by a simple biotransformation of olive leaf extract by a partially purified hyperthermophilic β-glycosidase immobilized on chitosan support [77].

#### 3.4.2. 2,2-diphenyl-1-picryl-hydrazyl-hydrate (DPPH) Method

The most used method for determination of antioxidant potential is 2,2-diphenyl-1-picryl-hydrazyl-hydrate free-radical method. This is an antioxidant assay based on electron-transfer which makes a violet solution in ethanol. This free radical is stable at room temperature. It can be reduced in the presence of an antioxidant molecule, giving rise to colorless ethanol solution. The use of the 2,2-diphenyl-1-picryl-hydrazyl-hydrate assay can be useful to estimate different products at a time because it provides a simple and fast way to evaluate antioxidants by spectrophotometry.

Usage of 2,2-diphenyl-1-picryl-hydrazyl-hydrate method was reported in studies, where the stability of oils which contain olive leaf extract was investigated [64,71].

Additionally, Bouaziz et al. (2010) reported that the free radical scavenging activities of the two extracts and pure compounds were in the following order: α-tocopherol > BHT > enzymatic hydrolysate extract > oleuropein > acetylated hydrolysate extract [76].

#### 3.4.3. 2,2′-azinobis-(3-ethylbenzothiazoline-6-sulfonic acid) (ABTS) Method

2,2′-azinobis-(3-ethylbenzothiazoline-6-sulfonic acid) method has the extra adaptability because it can be used at different pH levels (unlike 2,2-diphenyl-1-picryl-hydrazyl-hydrate, which is sensitive to pH lower than seven). This is valuable when studying the effect of pH on antioxidant activity of different compounds. Additionally, it can be used for measuring antioxidant activity of samples extracted in acidic solvents.

Furthermore, 2,2′-azinobis-(3-ethylbenzothiazoline-6-sulfonic acid) is soluble in aqueous and organic solvents so that it is valuable in estimating antioxidant activity of samples in various media and is currently most used in simulated serum ionic potential solution. The other advantage of 2,2′-azinobis-(3-ethylbenzothiazoline-6-sulfonic acid) method is that samples reacted rapidly with 2,2′-azinobis-(3-ethylbenzothiazoline-6-sulfonic acid) in the aqueous buffer solution reaching a steady state within 30 min.

The 2,2-diphenyl-1-picryl-hydrazyl-hydrate reacted very slowly with the samples. After 8 h the steady state was approached, but not reached. This slow reaction was also observed when 2,2′-azinobis-(3-ethylbenzothiazoline-6-sulfonic acid) reacted with samples in alcohol. This implies that the reactivity of the antioxidants in sorghums with these free radicals is slowed in alcoholic media.

Brahmi et al. (2013) reported using this method for investigating of antioxidant activity of *Olea europaea* leaves and fruit extracts collected in two different seasons [15]. Also, Brahmi et al. (2012) reported usage of this method in case of volatile fractions from three Tunisian cultivars of olive leaves [27].

#### 3.4.4. Ferric Ion Reducing Antioxidant Power (FRAP) Method

In this method, a ferric salt, Fe(III)(TPTZ)2Cl3 (TPTZ = 2,4,6-tripiridil-striazina), is used as an oxidant agent. The redox potential of this salt is approximately 0.70 V. It uses antioxidants as reductant employing an easily reduced oxidant system present in stoichiometric excess. At low pH values (3.6), reduction of ferric tripyridyl triazine (Fe III TPTZ) complex to ferrous form (which has an intense blue color) can be monitored by measuring the change in absorption [86].

#### 3.4.5. Primary and Secondary Oxidation Methods

Hydroperoxides are primary products of lipid oxidation. Following techniques have been developed to evaluate their formation: iodometric hydroperoxide measurement and ultraviolet measurement of conjugated dienes. On the other hand, a few assays that monitor the formation of secondary oxidation products are primarily based on spectrophotometric and chromatographic methods: thiobarbituric acid reactive substances (TBARS), aldehyde measurement by the anisidine assay, chromatographic measurement of volatile compounds, and formic acid measurement [87].

Bonoli-Carbognin et al. (2008) studied the oxidative stability of oil-in-water emulsions containing bovine serum albumin and virgin olive oil phenolic compounds. For determination of antioxidant potential, they used primary and secondary oxidation [88]. Bendini et al. (2006) during storage particularly observed continuous increment of primary and secondary oxidation products in the olive oil samples with lower phenolic content. The oxidation process was evaluated by measuring primary and secondary oxidation products [89].

## 4. Table Review of Phenolic Compounds Isolation Methods, Presence of Enzymes and Antimicrobial and Antioxidant Activity

In addition to their diversity, phenolic compounds, their isolation methods, presence of enzymes and investigated on microbes has widely been reported in scientific research, which are reviewed in Table 1.

## 5. Conclusions and Future Perspectives

Phenolic compounds (caffeic acid, verbascoside, oleuropein, luteolin 7-*O*-glucoside, rutin, apigenin 7-*O*-glucoside, luteolin 4′-*O*-glucoside, hydroxytyrosol) isolated from olive leaves have been shown to inhibit the growth of *Escherichia coli*, *Klebsiella pneumoniae* and *Staphylococcus aureus*.

Oleuropein, a main component isolated from olive leaves, has been shown to inhibit sporulation of *Bacillus cereus*. Hydroxytyrosol, a metabolite of oleuropein, was recently proved effective against clinical human pathogenic strains of *Salmonella typhimurium* and *Staphylococcus aureus*. It was concluded that tyrosol is a good inhibitor of bacteria Listeria monocytogenes and *Aspergillus flavus*. Rutin is good against bacteria *Pseudomonas aeruginosa* and fungi *Candida albicans* and *Candida neoformans*.

According to many reviewed studies, phenolic compounds from olive leaf extract have great antioxidant potential, especially oleuropein and hydroxytyrosol.

Oleuropein is extracted by different methods, in particular by solid-liquid extraction with ethanol as solvent. On the other hand, for hydroxytyrosol extraction, ethyl acetate is mostly used as a solvent.

Thermal treatment, especially discontinuous steam treatment, shows great potential for industrial application, and further investigation and improvements are necessary. Also, because of thermally labile and compounds which are low in volatility SFE may become a regular extraction technique for studying agricultural samples, food, and herbal.

## Figures and Tables

**Figure 1 molecules-25-05946-f001:**
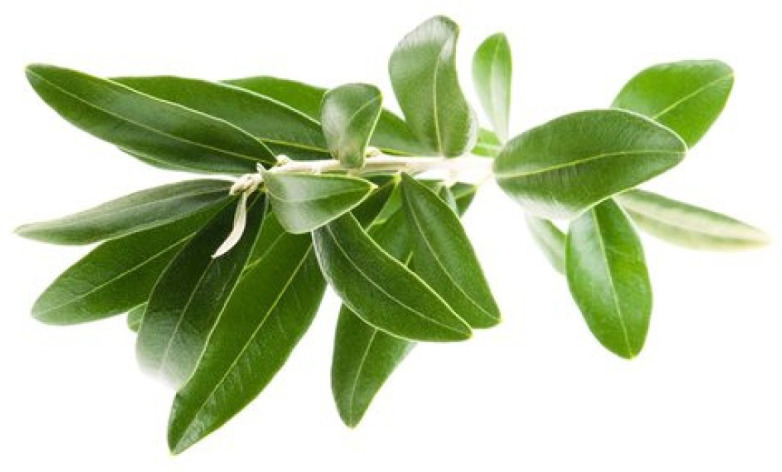
Olive (*Olea europaea* L.) leaves [5].

**Figure 2 molecules-25-05946-f002:**
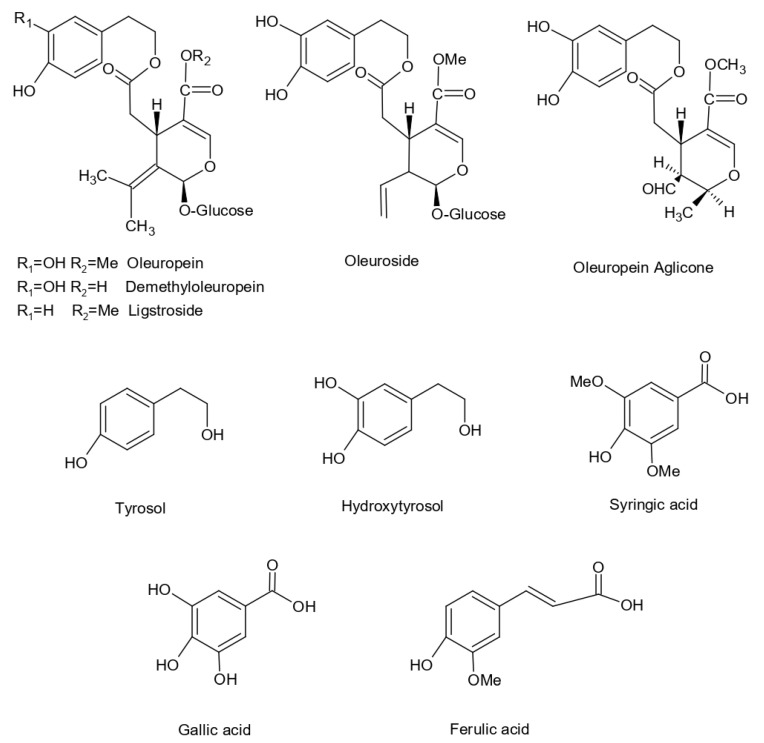
Chemical structures of phenolic compounds already identified in *Olea europaea* L. leaf extract.

**Figure 3 molecules-25-05946-f003:**
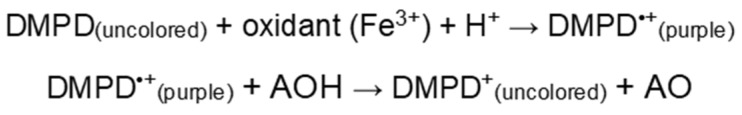
Scheme of *N*,*N*-Dimethyl-*p*-phenylenediamine dihydrochloride method [83].

**Table 1 molecules-25-05946-t001:** Review of isolation methods of phenolic compounds, presence of enzymes and antioxidant and antimicrobial activity against different microbes.

Extraction, Isolation	Compounds	Enzymes	Antioxidant	Microbes	References
Extraction with boiling H_2_O for 30 min	Caffeic acid, verbascoside, oleuropein, luteolin 7-*O*-glucoside, rutin, apigenin 7-*O*-glucoside, luteolin 4′-*O*-glucoside	/	/	*Bacillus cereus, Bacillus* *subtilis, Staphylococcus aureus* (Gram+), *Escherichia coli, Pseudomonas aeruginosa, Klebsiella pneumoniae* (Gram−) bacteria, and *Candida albicans* **and *C. neoformans* (fungi)	[6]
Extraction with H_2_O/EtOH (1:1, *v*/*v*) shaken for 15 min at 4 °C	Oleuropein, oleouroside, oleuropein aglycone, tyrosol, hydroxytyrosol, syringic acid, gallic acid, ferulic acid	+	DMPD method	/	[84]
Extraction with n-Hex and with EtAc	Hydroxytyrosol, tyrosol, hydroxytyrosol acetate, 3,4-DHPEA-EDA, oleuropein, 3,4-DHPEA-EA, 4-HPEA-EDA	/	/	/	[40]
Extraction with Ac/HCl and with EtAc	Oleuropein, caffeic acid, luteolin, luteolin-7-*O*-glucoside, apigenin-7-*O*-glucoside, quercetin and chryseriol	Inhibition of antioxidative enzyme activity	/	/	[80]
Extraction with boiled H_2_O for 15 min	/	Respiratory enzymes of bacterial cells	/	*Staphylococcus aureus*, *Pseudomonas aeruginosa* and *Escherichia coli*	[22]
Liquid/liquid extraction with EtAc	Oleuropein	Fungal enzyme (the β-glucosidase activity)	DPPH radical method	/	[90]
Extraction with H_2_O/MeOH (1:4, *v*/*v*) and left to stand overnight under agitation in the dark	BHT, hydroxytyrosol, hydrolysate extract, oleuropein, ethyl acetate extract and CH3OH/H_2_O leaf extract	The production of hydroxytyrosol by enzymatic hydrolysis	Using the β-carotene linoleate model system	/	[72]
Microwave-assisted extraction with magnetic stirring (6 min irradiation)	Phenolic compounds	/	Thermal oxidative stability analysis	/	[12]
Extraction with H_2_O/EtOH at different volume ratios	Oleuropein	Enzyme immobilization	DMPD method	/	[85]
Extraction with H_2_O/EtOH (3:7, *v*/*v*) for 2 h at 25 °C	Oleuropein, rutin	/	With aqueous ABTS solution	*Escherichia coli*, *Staphylococcus* *aureus*, *Klebsiella pneumoniae* and *Pseudomonas aeruginosa*	[24]
Extraction with H_2_O/EtOH (1:1, *v*/*v*) shaken for 15 min at 4 °C	Oleuropein, tyrosol, vanillic acid, hydroxytyrosol, 4-hydroxy, 3-methoxyphenyl acetic acid, 3,4-dihydroxy-benzoic acid, 3,4-dihydroxy phenyl acetic acid, syringic acid, gentisic acid, gallic acid, ferulic acid, caffeic acid, sinapic acid, oleuropein aglycon	Enzyme immobilization	DMPD method	/	[77]
Extraction with MeOH in a shaker at room temperature	Gallic acid, hydroxytyrosol, chlorogenic acid, protocatechuic acid, hydroxyphenylacetic acid, 4-hydroxybenzoic acid, catechin, oleuropeine, *p*-coumaric acid, ferrulic acid, rosmarinic acid, vanillic acid, m-coumaric acid, o-coumaric acid, phenylacetic acid, cinnamic acid, luteolin, apigenin, 3-hydroxybenzoic acid	/	DPPH radical scavenging assay; ABTS^+^ radical cation scavenging	/	[15]
Hydrodistillation for 3 h using a Clevenger-type apparatus	Furfural, (*E*,*Z*)-2,4-hexadienal, (*E*)-2-hexenol, (*E*)-3-hexenol, 1-hexanol, (*Z*)-4-heptenal, heptanal, (*E*,*E*)-2,4-hexadienal, 2-acetylfurane, α-Pinene, (*Z*)-2-heptenal, benzaldehyde, 3-ethenylpyridine, phenol, hexanoic acid, 3-octanone, 6-methyl-5-hepten-2-one, octanal, (*E*,*E*)-2,4-heptadienal, (*E*,*Z*)-2,4-heptadienal, benzyl alcohol, phenylacetaldehyde, (*E*)-2-octenal, 1-octanol, cis-linalool oxide, trans-linalool oxide, linalool, nonanal, phenylethyl alcohol, methyl nicotinate, 4-ketoisophorone, (*E*,*Z*)-2,6-nonadienal, (*E*)-2-nonenal, 1-nonanol, trans-linalool oxide (pyranoid), *p*-cymen-8-ol, α-terpineol, methyl salicylate, (*Z*)-4-decenal, decanal, 2-ethylbenzaldehyde, benzothiazole, geraniol, (*E*)-2-decenal, salicylic alcohol, *p*-menth-1-en-7-al, 1-tridecene, (*E*,*Z*)-2,4-decadienal, 4-vinylguaiacol, (*E*,*E*)-2,4-decadienal, eugenol, (*E*)-β-damascenone, cis-α-bergamotene, (*Z*,*E*)-2,6-dodecadienal, trans-α-bergamotene, (*E*)-isoeugenol, (*E*)-geranylacetone, (*E*)-β-ionone, caryophyllene oxide	/	DPPH radical scavenging assay; ABTS^+^ radical cation scavenging	*Enterococcus faecalis*, *Staphylococcus aureus*, *Escherichia coli* and *Pseudomonas aeruginosa*	[27]
Extraction with Ac by mechanical stirring for 12 h	Oleuropein, hydroxytyrosol	/	/	*Streptococcus mutans*, *Streptococcus sobrinus*, *Streptococcus oralis*, *Enterococcus faecalis*, *Candida albicans*, *Escherichia coli*, *Staphylococcus aureus*, *Porphyromonas gingivalis*, *Prevotella intermedia*, *Fusobacterium nucleatum* and *Parvimonas micra*	[25]
Extraction with 20% H_2_O and autoclaved for 20 min at 121 °C	Oleuropein, tyrosol, hydroxytyrosol, quercetin, *p*-hydroxybenzoic, vanillic, verbascoside and *p*-coumaric acids	Deglucosidation by the enzyme β-glucosidase to produce an aglycone structure of oleuropein	/	*Escherichia coli* and *Staphylococcus aureus*	[26]
Microwave-assisted extraction (8 min of microwave irradiation at 200 W) with H_2_O/EtOH (1:4, *v*/*v*)	Oleuropein and luteolin	/	/	/	[55]
Extraction twice with distilled H_2_O for 12 h at 80 °C	Rutin, verbascoside, luteolin7-glucoside, apigenin7-glucoside, flavonoid x, oleuropein and oleuroside	/	/	/	[23]
* High strength olive leaf extract was obtained from Spain	Oleuropein	/	/	/	[91]
Extraction with EtOH for 2 weeks at room temperature	Oleoside, hydroxytyrosol, tyrosol, aesculin, hydroxypinoresinol-glycoside, luteolin glucoside derivative, oleuropein and luteolin 7-glucoside	Plasma enzymatic activity; decreasing liver enzymes	/	/	[39]
Ultrasound-assisted extraction in an ultrasonic bath at 25 °C	/	/	DPPH radical method	/	[10]
* High strength olive leaf extract was purchased from a local health food store	Oleuropein	/	/	*Acinetobacter calcoaceticus*, *Bacillus cereus*, *Bacillus subtilis*, *Campylobacter jejuni*, *Candida albicans*, *Candida glabrata*, *Candida parapsilosis*, *Enterococcus faecalis*, *Escherichia coli*, *Helicobacter pylori*, *Klebsiella pneumoniae*, *Kocuria rhizophila*, *Lactobacillus acidophilus*, *Lactobacillus casei*, *Lactobacillus* spp, *Listeria innocua*, *Listeria monocytogenes*, *Micrococcus luteus*, *Pseudomonas aeruginosa*, *Salmonella enterica*, *Serratia marcescens*, MSSA, MRSA, *Staphylococcus capitis*, *Staphylococcus epidermidis, Staphylococcus hominis, Staphylococcus xylosus* and *Streptococcus pyogenes*	[3]
/	Oleuropein, tyrosol, hydoxytyrosol and caffeic acid	/	/	/	[81]
Extraction with H_2_O/EtOH (3:7, *v*/*v*) (was allowed to stand for at least one week at room temperature)	Apigenin 7-glucoside and oleuropein	/	/	/	[78]
Extraction in 20% H_2_O and autoclaved for 20 min at 121 °C	Oleuropein and hydroxytyrosol	/	/	Bacteria: *Escherichia coli*, *Pseudomonas aeruginosa*, *Staphylococcus aureus*, *Bacillus subtilis* and *Klebsiella pneumoniae*; Dermatophytes –*Trichophyton mentagrophytes*, *Microsporum canis* and *T*. *rubrum*; Yeast –*Candida albicans*	[28]
/	Oleuropein, oleuropein aglycone, elenolic acid and hydroxytyrosol	/	/	HIV-1	[92]
Extraction with EtAc	Vanillin, cinnamic acid, tyrosol, *p*-hydroxy-benzoic acid, *p*-hydroxy—phenylacetic acid, vannilic acid, hydroxy-tyrosol, protocatechuic acid, *p*-coumaric acid and ferulic acid	/	DPPH radical method	/	[71]
Extraction with H_2_O/EtOH (1:4, *v*/*v*)	Caffeic acid, vanilin, rutin, luteolin-7-*O*-glucoside, apigenin-7-*O*-glucoside, oleuropein, quercetin, luteolin, apigenin and chryseriol	/	DPPH radical method	/	[8]
Supercritical fluid extraction with CO_2_ and Soxhlet methods for 24 h	Oleuropein	/	/	/	[11]
Extraction with MeOH for 7 days in the dark at room temperature	Oleuropein, luteolin-7-*O*-glucoside, luteolin-4′-*O*-glucoside, luteolin and hydroxytyrosol acetate	/	DPPH radical method	/	[34]
Extraction with H_2_O/EtOH (1:4, *v*/*v*)	Oleuropein, caffeic acid, luteolin-7-*O*-glycoside, apigenine7-*O*-glycoside, quercetin and tannins	Antioxidative enzymes activity was compared with effects of i.g. pretreatment of reference drug, ranitidine	/	/	[4]
Extraction with H_2_O/EtOH (1:4, *v*/*v*) twice	Oleuropein, luteolin, apigenin, rutin, diosmetin, oleasterol, leine and glycoside oleoside	/	/	/	[93]
* Commercially available extract	Oleuropein, oleoside, hydroxytyrosol, luteolin-7-*O*-glucoside, tyrosol, verbascoside, apigenin-7-*O*-glucoside, rutin, vanillic acid, vanillin and luteolin	Inhibition angiotensin-converting enzyme in vitro and decreasing the activities of key cholesterol-regulatory enzymes	/	/	[94]
Extraction with H_2_O/EtOH (3:7, *v*/*v*) for 24 h at room temperature using a shaking incubator	Oleuropein	Antioxidant enzymes	/	/	[95]
Extraction with EtAc over-night at room temperature with constant stirring	Hydroxytyrosol, oleuropein, secoiridoids, flavonoids and triterpenes	/	/	/	[41]
* Standardized dry olive leaf extract was purchased	Oleuropein, luteoline-7-*O*-glycoside, apigenine-7-*O*-glycoside, quercetin, tannins and caffeic acid	/	/	/	[96]
Extraction with H_2_O/EtOH (1:4, *v*/*v*) twice	Oleuropein, tyrosol, hydroxy-tyrosol and caffeic acid	/	/	/	[97]
Extraction with distilled H_2_O in Soxhlet apparatus for 1 h at 60 °C	Oleuropein and flavonoids	Measuring of total activities of hippocampal enzymes, including glutathione-S-transferase and NADP-isocitrate dehydrogenase	/	/	[44]
Extraction with EtAc	Oleuropein and hydroxytyrosol	Enzymatic hydrolysis with different enzymes (β-glucosidase, hemicellulase, tannase, neutral protease, cellulase, glucoamylase, papain, alkaline protease, amylase, β-glucanase)	DPPH radical method	/	[13]
Extraction with H_2_O/EtOH (1:4, *v*/*v*)	Oleuropein, luteolin-7-*O*-glucoside, apigenine-7-*O*-glucoside, quercetin and caffeic acid	Antioxidant enzymes	/	/	[38]
Extraction with 70% EtOH for 24 h at room temperature by a shaking incubator	Oleuropein	/	/	/	[53]
Extraction with MeOH	Oleuropein	/	/	/	[70]
Extraction with H_2_O/EtOH (1:4, *v*/*v*)	Oleuropein, luteoline-7-*O*-glucoside, apigenine-7-*O*-glucoside, quercetin, tannins and caffeic acid	/	/	/	[98]
Extraction with H_2_O/EtOH carried out under magnetic stirring at 400 rpm and at room temperature (22 ± 2 °C) for predetermined time periods	Luteolin diglucoside, rutin (quercetin 3-*O*-rutinoside), luteolin glucoside, luteolin rutinoside, apigenin rutinoside and oleuropein	/	/	/	[75]
Extraction with H_2_O/MeOH and left to stand overnight under agitation in the dark	Oleuropein and hydroxytyrosol	Dehydrogenase enzyme	/	/	[37]
Quartz extraction with EtOH (8 min of microwave irradiation at 200 W)	Apigenin-7-glucoside, luteolin-7-glucoside and verbascoside	/	/	/	[82]
Extraction with H_2_O/EtOH (1:4, *v*/*v*)	Oleuropein, caffeic acid, hydroxytyrosol and tyrosol	Enzyme linked dimmunosorbent	/	/	[99]
Extraction with H_2_O/EtOH (1:4, *v*/*v*)	Oleuropein, hydroxytyrosol, caffeic acid, tyrosol, apigenin, apigenin-7-*O*-β-d-glucoside, luteolin-7-*O*-β-d-glucoside, luteolin and verbascoside	Catalyzed by the enzyme superoxide dismutase	/	/	[79]
Extraction with MeOH for 7 days in the dark at room temperature	Hydroxytyrosol glucoside, oleoside, hydroxytyrosol, secologanoside, oleuropein aglycon, 10-hydroxyoleuropein, verbascoside, hydroxytyrosol acetate, luteolin-7-*O*-rutinoside, 10-hydroxyoleuropein isomer, luteolin-7-*O*-b-d-glucopyranoside, oleuropein aglycon decarboxymethyl dialdehyde form, luteolin-40 -*O*-b-d-glucopyranoside, oleuropein, oleuropein aglucon, oleuropein isomer, oleuroside, ligstroside, luteolin and oleuropein aglycon	/	/	/	[73]
Extraction with different solvents (Ac, EtOH and their aqueous forms) for 24 h	Hydroxytyrosol, tyrosol, catechin, caffeic acid, vanillic acid, vanillin, rutin, luteolin-7-glucoside, verbascoside, apigenin-7-glucoside, diosmetin-7-glucoside, oleuropein and luteolin	/	ABTS/K_2_S_8_O_2_ method	/	[42]
Extraction with EtAc three times	Oleuropein, hydroxytyrosol and oleuropein aglycone	Enzymatic hydrolysis using β-glucosidase	ABTS assay	/	[16]
Extraction with H_2_O/MeOH and left to stand overnight under agitation in the dark	Oleuropein and hydroxytyrosol	/	DPPH radical method	/	[74]
Extraction with different solvents (H_2_O, EtOH, MeOH, CHCl_3_, CHCl_3_/EtOH and CHCl_3_/MeOH)	Hydroxytyrosol, tyrosol, catechin, caffeic acid, vanilic acid, vanilin, rutin, luteolin-7-glucoside, verbascoside, apigenin-7- glucoside, diosmetin-7- glucoside, oleuropein and luteolin	/	/	*Staphylococcus aureus*, *Escherichia coli*, *Salmonella enteritidis*, *Salmonella typhimurium* and *Listeria monocytogenes*	[7]
Extraction with absolute EtOH for 48 h	Caffeic acid, verbascoside, oleuropein, luteolin 7-*O*-glucoside, rutin, apigenin 7-*O*-glucoside and luteolin 4′-*O*-glucoside	/	/	*Escherichia coli*, *Staphylococcus aureus*, *Klebsiella pneumoniae*, *Bacillus cereus*, *Salmonella Typhi* and *Vibrio parahaemolyticus*	[29]
Extraction with H2O/EtOH in the dark at room temperature	Oleuropein, oleuropein aglycone, hydroxytyrosol and triacetylhydroxytyrosol	Enzymatic hydrolysis was carried out using β-glucosidase from almond	DPPH radical method	/	[76]
Microwave-assisted extraction with different solvents (MeOH, EtOH and their aqueous forms)	Oleuropein aglycone, luteolin diglucoside, luteolin glucoside, luteolin, quercetin, apigenin, apigenin-7-*O*-glucoside, oleuropein and rutin	/	/	/	[56]
Extraction with H_2_O/EtOH for 24 h at room temperature, by hydraulic laboratory press and supercritical-CO_2_ extraction	Hydroxytyrosol, vanillic acid, hydroxytyrosol glycoside, vanillic hexoside acid, caffeic hexoside acid, vanillin, oleoside, chlorogenic acid, oleuropein aglycon, pinoresinol, caffeic acid, elenolic acid, *p*-coumaric acid, ferulic acid, verbacoside, ligstroside aglycon decarboxymethyl, luteolin-0-rutinoside, acetoxypinoresinol, luteolin-7-glucoside, hesperitin-3-rutinoside, quercetin-3-0-galactoside, apigenin-7-rutinoside, oleuropein, oleuroside acid-10-carboxilic, apigenin-7-glucoside, oleuroside, ligstroside, luteolin-3′-7-diglucoside, luteolin-7-rutinoside, oleuropein diglucoside, oleuropein aglycon aldehyde and quercetin	Diphenol oxidase	DPPH method	/	[64]
Extraction with H_2_O/EtOH (1:4, *v*/*v*)	Oleuropein and verbascoside	/	/	*Listeria monocytogenes*, *Escherichia coli* and *Salmonella enteritidis*	[30]
Extraction with H_2_O/MeOH (3:7, *v*/*v*) in the dark for 48 h at 4 °C with periodic mixing	Tyrosol, oleuropein, caffeic acid, rutin, luteolin derivatives and vanillin	Superoxide dismutase and myeloperoxidase	DPPH radical method	*/*	[32]
Extraction with H_2_O/MeOH (1:4, *v/v*) and carried out by a household microwave (microwave irradiation was performed for 10 min)	Oleuropein, caffeic acid, tyrosol and hydroxytyrosol	/	/	*Aspergillus flavus*	[35]

* commercially.

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
