# Peer review of "Microbiological and Antioxidant Activity of Phenolic Compounds in Olive Leaf Extract"

_molecules, 2020, doi:10.3390/molecules25245946_

Round 1

Reviewer 1 Report

Since the authors have satisfactorily addressed all the questions proposed, I now believe that the manuscript can be published.

Reviewer 2 Report

All corrections and comments have been satisfactorily answered and included in the text, improving the quality of the work. The work is now suitable for publication.

This manuscript is a resubmission of an earlier submission. The following is a list of the peer review reports and author responses from that submission.

Round 1

Reviewer 1 Report

In my view the review by Borjan et al. need a very careful revisione and re-organization before it can be suitable for an eventual publication.

All my specific comments are listed in the attached file molecules-1010473-peer-review-v1_R1.

Author Response

Your comments have been addressed - throughout the manuscript the corrections have been marked with yellow color. Grammar and spelling mistakes have been addressed, thank you for corrections.

  1. Lines 17-22 Please, rephrase more clearly.

This part of paragraph has been rephrased.

  1. Line 32 Changed to “pharaons”.

Grammatically correct is with “h” as written previously.

  1. Lines 39-42 This information does not seem relevant.

This part of paragraph has been eliminated.

  1. Lines 46-47 What do you mean? Please, revise.

This part of paragraph has been changed.

  1. Lines 60-63 Pay attention, they are redundant with previous lines, for example lines 51-54. Therefore, decide wether group together or delete them.

This part of the paragraph has been eliminated.

  1. Figure 2 Please, improve the figure quality.

The quality of the picture has been improved.

  1. Lines 123-125 Move these lines to another more suitable paragraph.

Paragraph has been moved at the end of this section.

  1. Lines 133-139 Why did you choice to tell just on luteolin? What about the other cited flavonoids?

Luteolin was chosen because other cited flavonoids are its derivates.

  1. Lines 141-143 Move this lines to more suitable paragraph

The part of paragraph has been moved to Isolation methods.

  1. Lines 155-156 Please rephase

This part of sentence has been rephrased.

  1. Lines 206-207 Why should it be relevant to the antimicrobial activity?

This paragraph has been eliminated.

  1. Lines 214-214 What do you mean?

The sentence has been explained in a different way.

  1. Lines 243-244 Revise this sentence.

The sentence has been revised and changed.

  1. Line 246 Add suitable reference, such as: Crupi, P.; Dipalmo, T.; Clodoveo, M.L.; Toci, A.T.; Coletta, A. Seedless table grape residues as a source of polyphenols: comparison and optimization of non-conventional extraction techniques. Eur. Food Res. Technol. 2018, 244, 1091-1100.

The suggestion has been addressed, the reference has been added.

  1. Line 260 What do you mean? Are there SLEs which requires less than 24h?

Yes, but mostly it is between 24h and 48h.

  1. Lines 271-280 Be careful, yuo have already listed the typical solvents at lines 267-268.

This part of paragraph has been eliminated.

  1. Lines 306-309 Please, rephrase more clearly.

This paragraph has been rephrased.

  1. Lines 338-339 Please, revise.

This sentence has been eliminated.

  1. Lines 342-343 Rephrase.

This part of paragraph has been rephrased.

  1. Line 356 Are you sure? Which are the high equipment costs?

This part has been specified more precisely; it was meant to cost of solution consumption.

  1. Line 367 Move to line 357.

The sentence has been moved.

  1. Line 381 Be careful, it sounds a bit in contrast with line 259. Please, revise.

Precise duration time has been eliminated since it depends on several parameters and can therefore not be determined theoretically.

  1. Lines 383-385 I noted that microwave and ultrasound techniques should be better explained. Maybe,

the work "Clodoveo, M.L.; Dipalmo, T.; Rizzello, C.G.; Corbo, F.; Crupi, P. Emerging technology to develop novel red winemaking practices: an overview. Inn. Food Sci. Emerg. Technol. 2016, 38, 41 – 56" could be useful to the authors in this sense.

This part of paragraph has been precisely explained. Description has been written in accordance with suggested literature.

  1. Lines 353-385 Please, improve this explaination. It sounds quite unclear.

The explanation has been improved.

  1. Lines 409-411 Please, rephrase.

The sentence has been rephrased.

  1. Lines 416-418 Please, rephrase.

The sentence has been rephrased.

  1. Lines 420-422 Please, improve this explaination. It sounds quite unclear.

The explanation has been improved.

  1. Lines 427-442 Hard to follow this description, because you pass from talking about one type of

device to report the advantages of both type. Then you return to the first type. Please, revise.

The description has been reorganized and extended with same facts for both types (operating frequency, short process description, consumption of ultrasonic energy).

  1. Lines 437-440 Group and integrate with lines 420-422.

The part of paragraph has been moved as suggested and integrated with the rest of paragraph.

  1. Lines 440-442 Sorry, very unclear.

The sentence has been written in a different way.

  1. Line 460 Add a suitable reference.

Suitable references have been added.

  1. Line 471 Add a suitable reference.

The reference has been added.

  1. Lines 474-480 What does it have to do with the theme of the review?

Paragraph has been eliminated.

  1. Lines 481-492 Concepts expressed in these lines are similar to those reported previously. Therefore,

I would suggest to group them coherently.

Lines have been regrouped.

  1. Lines 493-561 In my opinion the concepts described in this paragraph are very confusing and often

not relevant. Please, revise and reduce.

Paragraphs have been adapted and reduced.

  1. Section 3.2 This section is very confusing. I would suggest to revise and re-organize it. Reading

the title of this review, I cannot understand what is the relevance of describing the analytical techniques? Instead of a poor description of GC, HPLC, NMR, and MS, I would suggest to group in this paragraph simply the literature findings about the polyphenols determination in olive leaves. Specify type of detectors.

The section has been reorganized and reduced as suggested. Types of detectors have been specified. Abbreviations of methods have been added.

  1. Lines 590-591 Please, rephrase.

The sentence has been rephrased.

  1. Line 603 From olive leaves?

Yes, it has been changed to “drive olive leaf extract”.

  1. Lines 713-715 Rephrase more clearly.

Sentence has been rephrased.

  1. Table 1 Edit the table more clearly.

Table has been arranged according to journal's template (Instruction for the authors). To be more clearly table has been oriented landscape (instead of portrait), font of letters has been reduced (from 10 to 9) as well as line spacing has been added.

  1. Section 5 Adjust conclusion, after revising the text. What about future persectives.

Conclusion has been revised. Thermal treatment and supercritical fluid extraction have been added as techniques with great potential for further improvement and application in industry.

Reviewer 2 Report

The article of review shows a good vision on the phenols obtained from the olive leaf, and focuses fundamentally and in greater depth in the microbiological aspects, reason why it must be improved as far as certain tactics very used for the extraction and in certain methods of determination of antirust activity that also are importants and lack in the present review. It is for this reason that the authors must improve the following aspects:
Line 13: change very good for important
Line 114: Change the expression “would probably decompose”, because the hydrolysis of the oleuropein because the light, high temperature, acid, or base has been demonstrated by many authors.
Line 124: The author should find some examples of synergist of phenols from olive, probably one of the most important is the synergist effect found with hydroxytyrosol and 3,4-dihydroxyphenylglycol, being both phenols together with the oleuropein the most actives compounds present in olive leaf.
Section: 3.1.2. Non-conventional extraction techniques: One of the most important technique that is starting to use in the olive oi industry is missing, the thermal treatments using high pressure (steam explosion) or low pressures (Steam treatment), both have been used to extract the main phenolics from all by-product from olive oil industry, including the leaf. These treatments use as the main heating agent the steam, as a difference of the supercritical or subcritical water. The author must be these kind of treatments in account.
Section: 3.4. Methods for determination of antioxidant activity: the main methods to measure the antioxidant activity as the radical scavenging activities in water has been described, but in lipid matrix like primary or secondary oxidation is missing. Other way to assess the antioxidant capacity is measuring the reduction power, that is widely used to study the antioxidant capacity of phenolics compounds in leaf extracts. All these methods must be in account for the author to complete the review.
Figure 1. It is too long and difficult to research, please find other format to compact as much as possible the table.
Section: Conclusion: The result obtained by the thermal treatment should be include in the manuscript and in this section, because its industrial potential.

Author Response

Your comments have been addressed - throughout the manuscript the corrections have been marked with blue color. Recommended aspects have been added, thank you for your suggestions.

  1. Line 13 change very good for important

It has been changed.

  1. Line 114 Change the expression “would probably decompose”, because the hydrolysis of the oleuropein because the light, high temperature, acid, or base has been demonstrated by many authors.

The expression has been changed.

  1. Line 124 The author should find some examples of synergist of phenols from olive, probably one of the most important is the synergist effect found with hydroxytyrosol and 3,4-dihydroxyphenylglycol, being both phenols together with the oleuropein the most actives compounds present in olive leaf.

Suggested examples have been added.

  1. Section 3.1.2 Non-conventional extraction techniques: One of the most important technique that is starting to use in the olive oi industry is missing, the thermal treatments using high pressure (steam explosion) or low pressures (Steam treatment), both have been used to extract the main phenolics from all by-product from olive oil industry, including the leaf. These treatments use as the main heating agent the steam, as a difference of the supercritical or subcritical water. The author must be these kind of treatments in account.

Treatments have been added according to suggestions.

  1. Section 3.4 Methods for determination of antioxidant activity: the main methods to measure the antioxidant activity as the radical scavenging activities in water has been described, but in lipid matrix like primary or secondary oxidation is missing. Other way to assess the antioxidant capacity is measuring the reduction power, that is widely used to study the antioxidant capacity of phenolics compounds in leaf extracts. All these methods must be in account for the author to complete the review.

ABTS as a redox method had already been described, and FRAP method has been added as well as primary and secondary oxidation methods [1,2].

  1. Table 1 It is too long and difficult to research, please find other format to compact as much as possible the table.

Table has been arranged according to journal's template (Instruction for the authors). To be more clearly table has been oriented landscape (instead of portrait), font of letters has been reduced (from 10 to 9) as well as line spacing has been added.

  1. Section 5 The result obtained by the thermal treatment should be include in the manuscript and in this section, because its industrial potential.

Thermal treatment has been introduced as technique with great potential for further improvement and application in industry.

  1. Lu, X.; Rasco, B.A. Determination of Antioxidant Content and Antioxidant Activity in Foods using Infrared Spectroscopy and Chemometrics: A Review. Critical Reviews in Food Science and Nutrition 2012, 52, 853–875, doi:10.1080/10408398.2010.511322.
  2. Miguel, M.G. Antioxidant activity of medicinal and aromatic plants. A review. Flavour Fragr. J. 2010, 25, 291–312, doi:10.1002/ffj.1961.